# A Comparative Cross-Sectional Study of the Prevalence and Determinants of Health Insurance Coverage in Nigeria and South Africa: A Multi-Country Analysis of Demographic Health Surveys

**DOI:** 10.3390/ijerph19031766

**Published:** 2022-02-04

**Authors:** Monica Ewomazino Akokuwebe, Erhabor Sunday Idemudia

**Affiliations:** Faculty of Humanities, North West University, Mmabatho 2745, South Africa; erhabor.idemudia@nwu.ac.za

**Keywords:** health, health demography, insurance coverage, Nigeria, South Africa, socio-demographic factors

## Abstract

***Background:*** The core Universal Health Coverage (UHC) objectives are to ensure universal access to healthcare services by reducing all forms of inequalities. However, financial constraints are major barriers to accessing healthcare, especially in countries such as Nigeria and South Africa. The findings of this study may aid in informing and communicating health policy to increase financial access to healthcare and its utilization in South Africa and Nigeria. Nigeria-South Africa bilateral relations in terms of politics, economics and trade are demonstrated in the justification of the study setting selection. The objectives were to estimate the prevalence of health insurance coverage, and to explore the socio-demographic factors associated with health insurance in South Africa and Nigeria. ***Methods:*** This was a cross-sectional study using the 2018 Nigeria Demographic Health Survey and the 2016 South Africa Demographic Health Survey. The 2018 Nigeria Demographic Health Survey data on 55,132 individuals and the 2016 South Africa Demographic Health Survey on 12,142 individuals were used to investigate the prevalence of health insurance associated with socio-demographic factors. Percentages, frequencies, Chi-square and multivariate logistic regression were e mployed, with a significance level of *p* < 0.05. ***Results:*** About 2.8% of the Nigerian population and 13.3% of the South African population were insured (Nigeria: males—3.4%, females—2.7% vs. South Africa: males—13.9%, females—12.8%). The multivariate logistic regression analyses showed that higher education was significantly more likely to be associated with health insurance, independent of other socio-demographic factors in Nigeria (Model I: OR: 1.43; 95% CI: 0.34–1.54, *p* < 0.05; Model II: OR: 1.34; 95% CI: 0.28–1.42, *p* < 0.05) and in South Africa (Model I: OR: 1.33; 95% CI: 0.16–1.66, *p* < 0.05; Model II: OR: 1.76; 95% CI: 0.34–1.82, *p* < 0.05). Respondents with a higher wealth index and who were employed were independently associated with health insurance uptake in Nigeria and South Africa (*p* < 0.001). Females were more likely to be insured (*p* < 0.001) than males in both countries, and education had a significant impact on the likelihood of health insurance uptake in high wealth index households among both male and females in Nigeria and South Africa. ***Conclusion:*** Health insurance coverage was low in both countries and independently associated with socio-demographic factors such as education, wealth and employment. There is a need for continuous sensitization, educational health interventions and employment opportunities for citizens of both countries to participate in the uptake of wide health insurance coverage.

## 1. Introduction/Background

In this contemporary era, societies are becoming increasingly diverse, with rapid changes, and they are being confronted with expanding inequalities and a growing need for accessibility of social facilities such as healthcare services. Thus, universal health coverage (UHC) has become a main concern in many developing and less developed nations, because UHC was introduced to provide equitable access to quality healthcare. Financial protection, which concerns medical expenses paid out of pocket, is an essential factor in accomplishing universal health coverage [1,2]. Studies have shown that considerable dependence on out-of-pocket payments (OOP) as the most important source of payment for medical needs not only has an inauspicious consequence on demand for services, but also represents an increasing financial problem among families, leading to deprivation [3,4]. In addition, existing studies have indicated that per capita income expenditure on health and welfare in many low- and middle-income countries is likely to increase quickly in the long run [5,6]. Moreover, the prevention of extremely harmful health disbursements and the necessity to strive towards universal health coverage have drawn the attention of governments and several stakeholders to provide health insurance schemes that will offer subsidized fees and healthcare accessibility to citizens, especially in developing countries such as Nigeria and South Africa [3,4].

In Nigeria, OOP payments contribute over 70% of the spending on health, significantly surpassing the recommended 30% threshold [7,8]. This percentage is among the highest in the world, and certainly the highest in Africa, bringing financial ruin to many family circles [7,8]. Nigeria has established remarkable but unspecified significant commitments to decreasing OOP in order to ensure increased access of citizens to quality primary healthcare services by ratifying into law the National Health Act in 2014. The National Health Act states that “*all Nigerians shall be entitled to a Basic Minimum Package of Health Services (BMPHS) funded by a basic healthcare provision fund (BHCPF) by deducting from the contributions of not less than 1% of the Consolidated Revenue Fund (CRF) of the Federal Government of Nigeria*” [2]. Pertaining to BHCPF expenditure formulated strategies, just about 50% of the basic healthcare provision fund is anticipated to go towards expansion and funding of the basic minimum package of health services that will influence instituting a State Contributory Health Insurance Scheme (SCHIS). Thus, the prospects and expectations of the BHCPF have stirred many states in Nigeria to set in motion the aim and execution of a bill on a State Social Health Insurance Scheme (SHIS). The Nigerian government has tried out many forms of SHIS in the past two decades. In 2000, the National Health Insurance Scheme (NHIS) was introduced with a coverage of about 4% of the entire population, most of whom were federal civil service employees [9,10]. However, the SHIS providers did not make many efforts to allow the easy participation of employed individuals within the informal sector to pay health insurance premiums. Several factors have been ascribed to the low coverage of SHIS, such as an absence of acceptability and refusal to pay premiums, specifically within the informal sector [10,11]. This is in agreement with previous studies, which showed that accomplishing economic sustainability and effective cross-subsidization through admission charges, particularly within the informal sector, are absolutely necessary contributing factors in implementing SHIS [12,13]. Health financing is key in closing the inequality gaps through instituting a health insurance system and ensuring accessibility to health services across the populations of countries [14,15]. There are strong indications of the need to implement a working health insurance system that will include employees within the formal and informal sectors, as an effective intervention to address OOP payments. Studies have cited that health insurance uptake has been extremely slow in Nigeria owing to the fact that NHIS funding is contributed to by the government and individuals who are working in government organizations [11,16]. In the post-independence era, health funding was largely from the government alone, in the form of free and universal healthcare, mainly in public facilities [17,18,19,20]. Now, the informal sector makes up about 70% of Nigeria’s labor force, and it is key to evaluate the practicability of the scheme’s funding through premium payments by looking into the Nigerian population’s willingness to enroll in the proposed scheme and pay premiums.

In South Africa, the government has made UHC one of the top priorities of the sustainable development goals to protect the vulnerable population from medical financial risks and to increase participation in the uptake of health insurance [4]. Despite progress made in certain aspects of the country after the end of apartheid in 1994, a well-established healthcare system is structured around a strict referral mechanism, yet wide disparities still continue to exist between the public and private health systems amid escalating medical costs and inequalities of underlying socio-economic factors, which have followed as a consequence of the apartheid regime [17,18]. Providing UHC for its citizens is to first recognize the significance of population health in countrywide advancement, and the government’s compelling public responsibility to make healthcare available to the citizens. In 2011, the South African government decided to institute a health insurance scheme that would provide subsidization of medical costs and reduce OOP payments with a single fund to cover all individuals, no matter their earnings [19].

However, there has been little determination and consideration to expand health insurance coverage in constrained resource-poor settings, which has become a serious public health concern in South Africa. Thus, about 8.5% of its gross domestic product (GDP) is spent on healthcare, with about half mostly spent on the private health sector, adequately providing for the elite group, despite the fact that about 84% of the remaining population carry a far greater disease burden, and who primarily depend upon the under-resourced public health sector [20]. As a result of this, efforts were made to introduce National Health Insurance (NIH) with the aim to accelerate its attainment in the national health policy agenda [3]. The NHI scheme was instituted to avert OOP payments for medical burdens, to provide financial risk protection to poorer households by reducing direct medical costs and to protect low-income households from OOP medical expenses and financial catastrophe related to healthcare services [2,3]. Several factors were cited in studies that found that even though health reforms were undertaken to integrate existing private schemes into NHI, medical aid providers were making huge profits from insurers and the majority of them were not enjoying a fair value for their premium contributions, especially those within the private sector. Moreover, many individuals were losing trust in public institutions and in the reformed NHI scheme, as fear exists that pooled resources from insurers’ contributions might not be used as intended for the masses.

A few studies have cited corruption in the health sector as one of the major barriers preventing pooled NHI funding from being publicly administered [3,21]. As regards to uptake of health insurance premiums, employment-based social health insurance premiums are restricted to the formal sector, which exclude most of the unemployed or employed individuals in the informal sector [22,23]. This has gradually made it difficult for individuals in the informal sector to have access to or procure a medical aid plan for themselves and their families. Employees are made to contribute up to two-thirds of their total monthly health insurance premium as part of tax deductible benefits [24,25,26]. Thus, the insured who can pay medical aid premiums have accessibility to first-world healthcare via the private sector, and those who cannot afford medical aid are left to rely on public hospitals, which are largely unreliable and fail to offer adequate medical attention to those without coverage and who pay a lot out of their pockets for medical bills [5]. Thus, health financing is not only designed to generate funds for healthcare delivery, but also to use pooled financial resources to support citizens who cannot afford medical aid premiums [27]. Hence, it is imperative to reflect on the background factors that may restrict or facilitate what can be implemented and achieved in South Africa [27,28].

Utilization of health insurance is ascertained by several underlying causes, comprising demographic, social and economic, as well as health status [24,25,26]. Understanding these factors that impede the uptake of health insurance will assist policy makers and relevant stakeholders to institute new effective interventions that will improve health insurance coverage to alleviate disease burden in the population. While there is a dearth of empirical studies on the comparative cross-sectional survey of the causes impelling health insurance coverage in Nigeria and South Africa, studies conducted at an individual country level have revealed that socio-demographic factors such as education, age, wealth status, residence, economic activities, religion, health status, parity, sex of head of the household, exposure to media, perception and willingness to become health insurance subscribers significantly influence health insurance coverage [29,30]. To bridge the present gap in understanding the comparative cross-sectional survey for a multi-country study through empirical observations, we examined the prevalence and predictors of health insurance coverage in Nigeria and South Africa, using the two study countries’ national demographic and health survey (DHS) data. The samples describe the countries’ populations in terms of key demographic characteristics of the entire populations in both countries. Based on the outcome variable, we employed multivariate logistic regression to attain suitable comparative study outcomes of the item-by-item contributing elements determining the total amount and type of health insurance accepted for its utilization in both countries. The study outcomes may perhaps impact or communicate this information or knowledge to policy makers and relevant health stakeholders regarding the factors which need careful deliberation when contriving as well as executing effective health insurance-related policy of intermediations carried out towards improving health outcomes by tackling poverty, increasing health insurance coverage and reducing the impoverishment accompanying payment for health services. This work also contributes to the World Health Organization (WHO)’s mission of the right to the highest achievable standard of health, “to Health for All and the SDGs” (SDG 3.8.1 and SDG 3.8.2), in achieving UHC in Nigeria and South Africa.

## 2. Methodology

### 2.1. Study Settings

This study uses data from the 2018 Nigeria Demographic Health Survey (NDHS) and the 2016 South Africa Demographic Health Survey (SADHS), which were conducted in each country. These countries have similarities and differences as far as geographical population structure and socio-economic environment are involved. The similarities and differences between Nigeria and South Africa are presented in Table 1.

These two countries were non-randomly selected to provide the comparative analysis of the geographical population distribution as well as differences in socio-economic disparities that are of concern to the study, taking into account the evaluation of their influence on health insurance coverage. Hence, the geographical distribution and socio-economic differences of two population groups are concomitant with large disparities in health resource distribution among different population groups, arising from social conditions [32,33].

### 2.2. Justification of the Study Settings

Nigeria and South Africa have been perceived as emerging giants on the African continent that have championed the repositioning of Africa on the route of long-term advancement and abridging her relegation in international economic relations. Presently, Nigeria enjoys economic power in the western part of Africa while South Africa enjoys economic supremacy in the southern part of Africa. Nigeria’s economy depends greatly on the oil sector, which contributes 95% of the country’s export revenue, while the South African economy is a diversified one, taking account of manufacturing, financial and mineral sectors, among others [34].

Thus, the justification of the selection of Nigeria and South Africa as the areas of study is evidenced in their many bilateral agreements and the establishment of the Nigeria–South Africa Bi-national Commission, which is committed to the consolidation and strengthening of political and social development, economic investment climates and trade relations, including bilateral relations in the fields of technology, education, health, culture, youth and sports [34,35].

### 2.3. Study Design

The present analysis is based on nationally representative datasets from two cross-sectional, population-based surveys conducted in Nigeria and South Africa. The 2018 NDHS was conducted in 2018 while the 2016 SADHS was conducted in 2016. The two Demographic Health Surveys (DHS) adopted a multi-stage cluster sampling methodology to arrive at the selected sample of adults aged 15–59 [36,37]. The two surveys used systematically sampled enumeration areas of the regions and districts to collect representative data for the age groups. The DHS was based on all persons in the household, but all the data used in this study were restricted to those aged 15–59 years. The DHS adopted a stratified two-stage probability sampling design to produce a sample representative of the target population. A detailed report on the methodology of the DHS conducted in Nigeria and South Africa has been presented elsewhere and cited in several studies using these demographic health surveys [10,11].

### 2.4. Data Sources

To draw inferences on the statistical distribution of the prevalence and socio-demographic determinants of health insurance coverage with its associated underlying factors, the study is based on nationally representative household surveys (2018 NDHS and 2016 SADHS) that provided data for a wide range of monitoring and impact evaluation indicators in the areas of population, health, and nutrition. Both surveys covered the following topics: the household, individual woman, individual man, caregiver, and biomarkers. The two surveys covered different age groups (women aged 15–49 years and men aged 15–59 years) and the survey analysis was restricted to adults aged 15–49 years old in order to make the analysis comparable between Nigeria and South Africa. Subsequently, the data used in the surveys represent only the sampled respondents; the data were weighted to make them nationally representative of the respondents aged 15–59 years. This was employed so that the design weights can be adjusted for household non-response and individual non-response to obtain the sampling weights for households and for women and men aged 15–49.

### 2.5. Sampling Procedures

A multistage stratified sampling technique was employed and carried out in the demographic health surveys in the two countries. Thus, the application of standard statistical methods for analyzing the data would yield standard errors that are not in conformity with the survey estimates, as the sampling methods used anticipated a simple random sampling technique. With respect to its inherent nature of data analysis, a multifaceted SPSS sample module was accepted to ensure that the standard errors for the survey estimates were expected to be robust to skewness and consistently reflect a precise and accurate criterion of sampling unpredictability. Thus, the sampling strategy adopted in the 2018 NDHS was a two-stage stratified design, where stage one engaged the collection of enumeration areas (EAs) using stratification proportional to size selection. Stage two instituted the chance of selection of about 30 households within each of the selected EAs [36,37]. From the 2018 NDHS, data were collected from 41,668 households of which 40,666 were occupied, providing in-depth individual data on 41,821 women and 13,311 men, adding up to 55,132 individuals and yielding a response rate of 99% for both men and women interviewed completely [37]. Furthermore, the sampling strategy employed in the 2016 SADHS was a stratified two-stage sample design, where stage one involved the selection of primary sampling units (PSU) using probability proportional to size selection [36]. Stage two entailed the systematic random selection of about 750 PSUs (primary sampling units) from 26 sampling household strata within each of the selected PSUs, and a systematic selection of 20 residential dwelling units (DUs) per cluster were selected. From the 2016 SADHS, data were collected on 15,292 households of which 13,288 were occupied, providing in-depth individual data on 8514 women and 3618 men. The sampled PSUs yielded a sample of 12,132 men and women interviewed completely with response rates of 86% and 73%, respectively [36].

### 2.6. Measures

#### 2.6.1. Outcome Variables

The dependent variable in this study is health insurance coverage. The households were classified into two groups based on individuals’ responses regarding whether they were insured or covered by any health insurance. Those who said they were insured or covered were coded as 1, and those who said they were uninsured or not covered by any health insurance were coded as 0.

#### 2.6.2. Independent Variables

The independent variables used for analysis in this study were selected on the basis of a literature review and on the availability of the limited socio-demographic variables collected by the 2018 NDHS and 2016 SADHS. The following variables were collected in all the surveys: gender, age, education, place of residence, region, province, race, wealth index, marital status, and employment status. Age was recoded in ten-year groups and categorized as follows: 15–24, 25–34, 35–44 and 45–59. The education variable was placed into four categories: no education, primary (including primary incomplete and primary complete), secondary (including secondary incomplete and secondary complete) and higher (more than secondary). However, in both surveys, the educational level reflects the highest educational level that the respondent attended [38], but does not inevitably mean that the level of education was completed. The wealth index was coded into three categories: poor (including lowest and second), middle, and rich (including fourth and highest). The wealth quintile variable is based on a wealth index factor score created using principal components analysis derived from data collected about a household’s cumulative living standard [38,39]. Therefore, the wealth quintile is a composite measure of a household’s cumulative living standard. Marital status was recoded into three categories as follows: never married, currently married (including married or living together) and previously married (divorced, separated or widowed). Employment status was classified as unemployed or employed. It is imperative to mention that the surveys have a few variables that were described as similar or equal, such as region/province and urban/rural or urban/non-urban, except for population group. This is peculiar to the South African population (Black African and “Others”—White, Colored, Indian/Asian and Other) and was recoded into two categories, and included as an important variable in the analyses. Other key socio-demographic factors such as religion and ethnic group were not used in the analyses owing to the fact that these variables were low on the list of main concerns as informed by the users of census data [40]. Therefore, to maintain consistency across surveys, and with the inclusion of population group, only the variables present in both surveys were used.

### 2.7. Statistical Analysis

The data were weighted using sample weights to adjust for degree of differences in probability selections, as the sample design involves more than one stage of selection. This ensured that data were cleaned and are representative of the target population, in this case, those aged 15–59 years. Descriptive analyses were first completed to define the characteristics of the study respondents. Proportions were calculated to depict the prevalence of health insurance ownership by country and gender. Pearson’s chi-squared test (χ^2^) was employed to examine the association between health insurance coverage and predictor variables (socio-demographic factors). Multivariate logistic regression analysis was used to ascertain the factors connected with health insurance coverage. Unadjusted Model 1 and adjusted Model 2 were constructed in the logistic binary regression, and only socio-demographic variables were included to bring out relevant findings for this study. Hence, confidence intervals (CIs) were applied to indicate the accuracy of the odds ratios (ORs); ORs and 95% CIs after unadjusted Model 1 and adjusted Model 2 for covariates were estimated and presented. All analyses were carried out using the STATA 17 software package (StataCorp: College Station, TX, USA).

### 2.8. Ethical Considerations

All data were obtained from the 2018 NDHS and 2016 SADHS. Informed consent was obtained from each respondent before the interviews. We obtained approval to use the data from the DHS repository (http://dhsprogram.com/data/available-datasets.cfm accessed on 24 May 2021).

### 2.9. Patient and Public Involvement

The study used secondary data from DHS and therefore, the dependent and independent variables used in the study were those already in existence in the datasets. Hence, no patients were included in the design or development of the research question and outcome measures. The findings will be disseminated to study respondents through the preparation of policy briefs and presentations in symposiums.

## 3. Results

### 3.1. Demographic Characteristics of the Respondents

Table 2 presents the distribution of the respondents by demographic characteristics. The proportion of the study respondents by gender indicates that there were more females (Nigeria—71.3%; South Africa—51.6%) than males (Nigeria—28.7%; South Africa—48.4%) who participated in the survey. The findings revealed that respondents between the ages of 15 and 24 years constituted 35% in Nigeria and 34.8% in South Africa. The majority of the respondents were educated to the secondary level in South Africa (75.2%) compared to Nigeria (41.5%), while more respondents were educated to the tertiary level in Nigeria (12.0%) compared to South Africa (10.0%).

More than half of the Nigerian respondents (59.2%) were found to be residing in rural areas compared to South Africans (56.3%), who were mostly found residing in urban areas. In South Africa, a majority of the respondents were Black Africans (86.6%) compared to the “Other” population groups (13.4%) who participated in this study. A majority of the respondents in South Africa were in the poor wealth quintile (43.1%) while 40.7% of the respondents in Nigeria were in the rich wealth quintile. As regards to employment status, most of the respondents in Nigeria were working (70.1%) compared to 36.2% of South Africans who had employment (Table 2).

### 3.2. Distribution of Health Insurance Coverage by Country

Overall, we found that more than 50% of these DHS populations have no health insurance coverage. From these study findings, the prevalence of health insurance coverage was 2.8% in Nigeria and 13.3% in South Africa. This finding further revealed that a higher proportion of respondents (97.2%) were not insured in Nigeria compared to uninsured respondents in South Africa (86.7%), as this infers inequalities in health insurance coverage in both countries (Figure 1).

### 3.3. Distribution of Health Insurance Status by Gender in Nigeria and South Africa

Figure 2 presents the distribution of respondents by health insurance status by gender in Nigeria and South Africa. These study findings reveal that the proportion of male respondents who had health insurance coverage was higher in both countries (Nigeria—3.4% and South Africa—13.9%) compared to female respondents (Nigeria—2.7% and South Africa—12.8%). However, health insurance ownership was much lower among both the male and female respondents in Nigeria compared to those in South Africa. Female respondents in Nigeria had the lowest proportion of health insurance coverage (2.7%) compared to females in South Africa (12.8%).

### 3.4. Distribution of Respondents by Health Insurance Status and Socio-Demographic Factors

Table 3 presents the distribution of respondents by health insurance status and socio-demographic factors. Apart from the gender variable, the Chi-square test revealed a significant association between socio-demographic factors (age, education, place of residence, region, province, race, wealth quintile, marital status and employment status) and respondents’ health insurance status in both Nigeria and South Africa (*p* < 0.05).

### 3.5. Determinants of Health Insurance Coverage

Table 4 presents the findings of the binary logistic regression analysis for the variations in health insurance coverage in Nigeria and South Africa among male and female respondents. In Models I and II, the study results showed that female respondents were more likely to have health insurance coverage than their male counterparts in both countries (*p* < 0.05). In only the adjusted Model II, the respondents’ age being 25–59 years was found to be associated with increased odds of having health insurance compared to those aged 15–24 years in South Africa (*p* < 0.05). In Nigeria, respondents aged 15–59 years were found to have lower odds of having health insurance coverage in both models (*p* < 0.05). Respondents with tertiary education in Nigeria were 14% (Model I) and 13% (Model II) more likely to have health insurance coverage compared to those with no education (uOR 1.43; 95% CI 0.34 to 0.54, *p* < 0.05; aOR 1.34; 95% CI 0.28 to 0.42, *p* < 0.05). In South Africa, respondents with tertiary education were 13% (Model I) and 18% (Model II) more likely to have health insurance coverage compared to those with no education (uOR 1.33; 95% CI 0.16 to 0.66, *p* < 0.05; aOR 1.76; 95% CI 0.34 to 1.72, *p* < 0.05). In Nigeria, rural respondents had 32% (Model I) and 11% (Model II) higher odds of having health insurance coverage compared to those in urban areas (uOR 3.23; 95% CI 2.89 to 3.63, *p* < 0.05; aOR 1.07; 95% CI 0.94 to 1.23). In South Africa, Model I showed that rural respondents had 25% higher odds of having health insurance than those in an urban place of residence (uOR 2.52; 95% CI 2.15 to 2.95, *p* < 0.05), whereas Model II showed that rural respondents had 6% decreased likelihood of having health insurance coverage than those in an urban residence (aOR 0.60; 95% CI 0.47 to 0.77, *p* < 0.05).

Regarding region and provinces, male and female respondents from regions other than South West and North West regions of Nigeria had higher odds of being insured compared to those from the North Central region (*p* < 0.05). In the uOR of Model I, male and female respondents in all the provinces in South Africa had higher odds of being insured compared to those from Western Cape, while in the aOR of Model II, those in provinces such as Eastern Cape, North West, Gauteng and Limpopo had lower odds of being insured. The likelihood of health insurance uptake increased with wealth status, as respondents in the rich wealth quintile had more likelihood of being insured than those in the poor wealth quintile in both countries. Moreover, working respondents and those who were previously married had a higher likelihood of having health insurance coverage in both Nigeria and South Africa. By population group, the “Other” population group had an increased likelihood of being insured than the Black African population group in both models.

## 4. Discussion

We assessed the prevalence and determinants connected with health insurance uptake among men and women in Nigeria and South Africa. Our results showed that the prevalence of coverage of health insurance in South Africa was the highest (13.3%), while that of Nigeria was the lowest (2.8%). This stems from the fact that the level of importance attached to healthcare and health insurance financing in South Africa is higher than that in Nigeria. Countries such as Ghana, Kenya, Tanzania and Uganda have a high coverage of health insurance, as cited by a few studies [41,42]. For instance, with an estimated population of 59,620,000, South Africa’s public expenditure on health in the year 2020 was ZAR 58.4 billion of the country’s total public expenditure [43]. With an estimated population of 206,139,589, Nigeria has 3.75% of its public expenditure out of NGN 495 billion GDP going to the health sector (ZAR 7.4 million) [44]. This partly explains why coverage of healthcare and insurance coverage was higher in South Africa [43] and lower in Nigeria [44,45], leaving a huge proportion of the population potentially disadvantaged when accessing healthcare services. Moreover, South Africa’s higher coverage may be ascribed to the coordinated and combined public and private health insurance scheme, ensuring risk pooling and increasing the confidence of potential subscribers or insurers in the health insurance system, hence encouraging them to subscribe [25,26]. Furthermore, the scheme is decidedly made to cover working individuals in government and private organizations, and non-working individuals can also have access to other lower medical aid plans, which makes it likely for all impoverished individuals to subscribe without paying other fees for the required annual premiums [29,46]. Financial contributions to the National Health Insurance Department (NHID) in South Africa are designed in such a manner that premium payments are graded according to people’s wealth status and ability to pay; individuals with a higher income are made to pay higher premiums compared with those with a lower income. Particularly, even though South Africa recorded higher coverage in this study, it is still far from the universal health coverage target (80%^+^), which is be achieved with the transition from the Millennium Development Goals (MDGs) to SDGs [41,46].

Contrary to the coordinated and combined health insurance system in South Africa, where there are public and private health insurance schemes, the health insurance program of Nigeria is a scheme public funded by government and insurers’ contributions, which are extremely uneven. This might likely and adversely affect resource pooling and cause inefficiency on the part of the scheme’s providers, which can create a sense of uncertainty about the scheme’s benefits among potential subscribers. This may prevent people from subscribing to the schemes, especially in Nigeria, which recorded one of the lowest coverage rates in Africa [39,47]. The low health insurance coverage in Nigeria could be attributed to cumbersome claiming processes and poor-quality services provided in accredited health facilities [14,44]. In terms of healthcare, African governments are facing a number of challenges, including lack of funds and poor infrastructure as well as corrupt practices within the health sector [21,48]. Nigeria has a public health service financed through a national insurance scheme, yet it faces a number of difficulties, including a low ratio of doctors to patients, even on a global scale, and an infrastructure struggling to cope. This is compounded by epidemics, poverty and the brain drain of homegrown doctors moving abroad in search of higher wages and a better standard of living [49,50]. Thus, strengthening health financing will assist indigents to access quality healthcare services without paying out-of-pocket for their medical treatment.

Our findings also showed that the prevalence of health insurance coverage by gender was 3.4% of men and 2.7% of women insured in Nigeria, while 13.9% men and 12.8% of women were insured in South Africa. In South Africa, these findings corroborate evidence from the 2016 South Africa Demographic Health Survey report on health insurance coverage by gender [36]. However, the NDHS [37] reported a slight increase in health insurance coverage among men (3.0%) than women (3.0%) compared to this study’s findings. The gender variable was a significant predictor of health insurance coverage and the likelihood of having health insurance coverage was higher for male respondents in Nigeria and South Africa. Previous studies have shown that female respondents have higher levels of support for health insurance coverage than men [51]. Thus, females are considered to be more active users of the health system compared to males [52]. Contrary to this presumption for females given previous studies’ findings [53,54], the present study found that males have higher odds of having health insurance coverage than females.

The direction of influence of gender on health insurance uptake is varied in the literature; one study did not find significant differences in health insurance coverage based on sex in Kenya [55,56,57]. Conversely, few studies have shown that decreased insurance uptake among males is a result of them seeming to be risk-takers [52,53], while further studies have reported that increased health insurance uptake by women was as a result of greater needs for healthcare services [58,59]. However, the decrease in women’s uptake of health insurance in both countries may be associated with the low socio-economic status of women relative to men. Women are more economically disadvantaged, and typically have poor access to health programmes owing to the extremely patriarchal pattern occurring in rural communities [58,59,60]. Furthermore, women’s education is lower, ensuing lower participation rates in social and health interventions, inferring that women may likely seek alternative healthcare services away from the orthodox health system.

The findings of the bivariate Chi-square test showed that the likelihood of respondents’ health insurance uptake was significantly increased by socio-demographic factors (age, education, place of residence, region, province, race, wealth index, marital status and employment status) in both countries (*p* < 0.05). Similarly, the multivariate analysis findings revealed that the chances of having insurance coverage are significantly increased by certain socio-demographic determinants in Nigeria and South Africa (*p* < 0.05). The findings showed that gender has a positive impact on demand for health insurance (*p* < 0.05). The likelihood of having health insurance coverage is 13% and 19% greater for females compared to males in Nigeria in Models I and II, respectively. In South Africa, the likelihood of females having health insurance coverage was 11% and 19% greater than males in Models I and II, respectively. Thus, females were more likely to enroll to become beneficiaries compared to males in both countries. This is similar to previous studies that reported that women of reproductive age were more likely to obtain more information on health insurance than men [58,59]. Even though women were more likely to lack a support system and have poor participation in economic activities to purchase health insurance plans, studies have reported that women were more willing to pay to become active health insurance subscribers [58,59]. The association between gender and health insurance is similar but more complex in other research. Studies conducted in Central Malawi, North-West Cameroon and Ghana identified males to have lower odds of being insured than females [53,54,61]. Other studies have also documented that women, as caregivers, are more conscious of the importance of health insurance, and are more likely to seek healthcare for themselves as well as for their families [59]. Even though several studies have cited that women were more likely to be insured than men, these studies could not establish a plausible explanation for this observation. It is important that future studies explore this inference.

The effect of age is positive, but a lower chance of having health insurance coverage was found among respondents aged 25–59 years in Nigeria compared to those aged 15–24 (*p* < 0.05) in both models. This could agree with previous studies that reported that a substantial proportion of individuals are unwilling to contribute to health insurance premiums, as they do not attach any significance to it [29,30]. In South Africa, respondents aged 25–34 years, 35–44 years and 45–59 years were found to be 18%, 13% and 10% more likely to have health insurance coverage compared to the younger age group 15–24 (*p* < 0.05). This could be because younger age groups are mostly dependents and beneficiaries of health insurance coverage purchased by their “significant others” (in sociology, a “significant other” describes any person(s) with a strong influence on an individual’s self-concept such as parents, close friends, spouse, siblings, etc.), as they are living with others who will include them in their insurance coverage [2,47]. Regarding age, respondents aged 35–59 years are perceived to be energetic and employed, as they are more concerned with good health conditions and have a strong sense of purpose for life. They are likely to engage in sporting activities at this stage of their lives, and adopt healthy behaviors in order to avoid declining health status, so they will opt for health insurance coverage. Previous studies have indicated that older persons might tend to increase their participation as health insurance subscribers compared to younger age groups [47,62]. Consistent with earlier studies, our finding established the possibility of having health insurance coverage rise with increasing age [14,32,63]. One likely justification for this finding is that financial security increases with age, which in turn enhances health insurance acquisition [8,14].

As anticipated, education increases the probability of taking up insurance of all types, with more educated individuals intending to have health insurance coverage, as education plays an essential role in the levels of awareness of health insurance schemes. The odds of having health insurance coverage among respondents with tertiary education were found to be greater than those who had no education in both the unadjusted and adjusted logistic regression models, and this finding was significant (*p* < 0.05). The likelihood of having health insurance coverage for those with tertiary education compared to primary and secondary education was highest in the multivariate analysis. Studies have shown that there is a directly proportional relationship between education level and subscribing to a health insurance scheme [1,64]. The implication of this study finding is that education has the ability to expose one to information in strategic discussions that will increase sensitization toward health insurance benefits in both countries. A similar study confirmed that higher education is associated with an individuals’ increased level of knowledge and perception toward short-and long-term benefits of health insurance. Thus, educated individuals have the capacity, not only to acquire skills and knowledge, but also to make informed choices on health-related matters in order to avoid catastrophic health expenditures [65,66].

Regarding place of residence, rural Nigerian respondents had 32% (Model I) and 11% (Model II) higher odds of having health insurance coverage than their urban counterparts. In South Africa, rural respondents had 25% (Model I) and 6% (Model I) higher odds of having health insurance compared with those in urban areas. The variations associated with rural and urban residence might explain the likelihood of rural residents to opt for shared legal health insurance schemes, with tendencies in South Africa to come together in social self-help groups to purchase health insurance coverage [67,68]. However, in rural areas of residence, individuals with poor self-assessed health status are more likely to be subscribed to health insurance by their significant others who reside in the urban cities [69]. This suggests that individuals with poor health in rural communities in both countries will self-select into health insurance schemes. Hence, in understanding the fundamental principles and underpinnings surrounding rural residents with health insurance ownership which are different from their urban counterparts, pragmatic aspects such as adverse selection, risk aversion, affordability and trust in health insurance plans can give better explanations and put these findings into proper perspective, but these aspects are beyond the scope of this study.

Region of residence was also a significant predictor of health insurance ownership. To be precise, men and women residing in the geographical regions of North West, South East, South South and South West had increased odds of having health insurance compared to the North Central region in Nigeria. The geographical differences in health insurance coverage could be explained as most of these geographical regions are almost completely urban and have a greater percentage of the population with rich status and higher literacy levels compared with the other geographical regions [1,37]. This outcome is consistent with earlier studies, which indicated that urban regions had increased odds of being insured [1,59]. In South Africa, province was found to be a significant predictor of health insurance. To be precise, in Model I, men and women residing in all the other provinces have increased odds of being insured compared to those from Western Cape, while in Model II, men and women residing in Eastern Cape, North West, Gauteng and Limpopo have lower odds of health insurance uptake. This outcome is consistent with earlier studies, which revealed that a larger proportion of the population was engaged in economic activities that might propel them to purchase a health insurance plan [37,56].

Race was also a significant predictor of health insurance coverage, with White, Colored and Indian/Asian respondents having higher odds of health insurance coverage compared to Black Africans, even though most users of the public health system in South Africa are Black Africans, as is most of the population [70,71]. This finding is consistent with other studies that found that the “Other” population groups (White, Colored and Indian/Asian) tend to have more investments in healthcare and therefore have a greater likelihood of health insurance coverage compared to Black Africans. Few studies have reported that “Other” population groups’ accessibility to health information and medical aid benefits has influenced their perceptions towards health insurance ownership [58,72]. In Nigeria and South Africa, health insurance schemes are designed to address health inequalities and ease the financial burden on health expenditures, yet a majority of health insurance subscribers were mostly found in the wealthier quintiles compared to those in the poor wealth quintile [1,73]. Moreover, household wealth status was also an essential contributing factor, as the likelihood of being insured increased as one moved up the household wealth index. This finding is consistent with earlier studies which indicated that wealthier households have increased odds of being insured as they can afford the health insurance plans [41,73]. Poor households with the likelihood of financial challenges in the future are less likely to sacrifice their current earnings and contribute to health insurance coverage to reduce future health risks [41,74].

Regarding employment status, our findings suggest that employed respondents were more likely to have health insurance in both countries. These findings are indicative of the fact that poor and unemployed persons have a limited ability to pay the regular premiums for health insurance. Our findings are consistent with comparable studies, which have shown that most unemployed individuals rely heavily on out-of-pocket (OOP) payments for health costs [69,75] and self-medication when they need medical attention [76,77]. The high coverage of health insurance among employed persons may be attributed to workplace insurance policies in government and large private organizations in Nigeria and South Africa. This implies that the governments of both countries should create platforms for formal employment opportunities for their citizens in order to increase health insurance coverage, especially in informal settlements or rural locations [55,78]. The study findings revealed that about 30% and 66% of unemployed respondents have no health insurance coverage in Nigeria and South Africa, respectively. This infers that health insurance uptake by unemployed persons in the informal sector is likely to be considerably lower, with over 30% of the study population likely be involved in menial jobs in the informal sector. Health insurance is mostly subscribed to by government employees [79,80,81,82,83].

The Nigerian dependency ratio is 86.7% and the South African dependency ratio is 52.2%. This will possibly put further pressure on the active population who are unemployed but carrying out day-to-day activities in the formal sector to meet with their daily needs [82,83]. It is therefore pertinent that stakeholders should employ subsidization and adjustable mechanisms of insurance uptake by those in the informal sector by working with micro-finance banks and the informal sector, which is being piloted across some specific states in Nigeria [84,85]. Hence, governments should also explore new opportunities to subsidize premiums to allow the non-working population to enroll in the scheme without facing further financial hardship in Nigeria and South Africa. This may take account of individual donor support and government matching subsidies on contributed premiums, which is comparable to the mechanism experimented in Tanzania where the health insurance scheme was partly funded by government, contributing towards achieving higher coverage rates in the country [59,86].

The study findings also show that marital status is a significant factor in explaining having health insurance coverage, and this variable has not been given much consideration in many studies conducted on factors that influence individuals’ decision to enroll in health insurance [59]. It is observed that being currently married is positively related to a higher likelihood of enrolling in health insurance for both male and female respondents compared to never being married or previously married. For instance, this study revealed that respondents who were previously married had reduced odds of having health insurance coverage by 0.74 and 0.74 in Nigeria, and 0.38 and 0.53 in South Africa in Model I and II respectively, compared to those who were never married. It could be inferred that married persons may take better care of themselves since they have significant others who are depending on them. There is limited evidence in the association between an individual’s marital status and health insurance coverage. Yet similar to previous studies, married individuals were more likely to have health insurance coverage than those who were not married or previously married [59,87]. A possible explanation could be that married persons may need to protect themselves and their families from unexpected health disbursements and out-of-pocket payments.

Emphasis on demographic and socioeconomic factors as significant predictors of health insurance coverage in this study is important, as these factors have been identified as pathways to increase health insurance coverage in line with the World Health Organization’s mission and mandates of achieving universal health coverage in 2025 [5]. Thus, these demographic and socioeconomic indicators are of particular importance in Nigeria and South Africa, as both countries are striving to achieve the UHC 2025 mandate with their involvement in substantial investments in health financing. Significantly, these indicators presented in the study findings for both countries are a primary interest of research that have important implications for individual and aggregate human behavior, having an impact on population health [35,36]. These aforementioned assumptions of the study findings are a reflection of the social determinants of the health profile and quality of healthcare services both countries provides for their citizens through health insurance coverage [88,89]. Nevertheless, policy makers should employ the use of demographic and socioeconomic statistical information to create public policy options for scaling up health insurance coverage in both countries that will lead to the attainment and sustainability of health insurance coverage in Nigeria and in South Africa.

However, several studies have shown that health insurance has generated inequities and contributed to healthcare inefficiency in South Africa [71,90]. These include providing access only to formal employees, excluding informal employees from being insured. Such inequalities have been a major concern following the apartheid era. Government and large private organization employees have been given more access to private health services, rather than improving access and quality of public health services for all South Africans. Thus, several studies have submitted that it is unlikely for the scheme to be undone or undergo a major reconfiguration, as inequities have long existed in South Africa [3,90]. On the other hand, the Nigerian government made some accelerations towards decentralizing universal health coverage to individual states in 2014 via the National Health Insurance Scheme (NHIS). Yet in 2018, about 97% of Nigerians did not have any health insurance and only about 3% of Nigerians with health insurance were under employer-based coverage [1,36], as privately purchased insurance plans were conspicuously uncommon in Nigeria. Presently, the Nigerian healthcare system is undergoing major reforms aimed at achieving universal health coverage, as the federal government has directed all state governments to set up and run mandatory state health insurance schemes (SHIS). Thus, with the passage of the bills, the National Health Insurance Scheme (NHIS) has the mandate to protect all Nigerians from paying for healthcare out-of-pocket, if effectively implemented, monitored and evaluated by addressing all forms of fraud and corruption by all relevant stakeholders [36,91].

Our findings have two significant policy implications in this multi-country setting. First, the low prevalence of health insurance coverage in Nigeria and South Africa highlights the urgent need for governments to take heed of the Kenya National Hospital Insurance Fund (NHIF) Strategic Plan Health Insurance Scale-Up to decrease healthcare financing burdens, especially among women, the unemployed and poor households. The explanations for this first policy implication in this study is rooted in the historical dimensions of the Kenya NHIF and how the government has successfully implemented and increased universal health insurance coverage in Kenya [92]. Previously, access to quality healthcare was a constitutional right, yet millions of Kenyans could not afford to pay for health services at public or private clinics and a quarter of total spending on healthcare came from out-of-pocket expenses [93]. However, that is changing as the collaborative Health Insurance Subsidy Program (HISP) was launched by the Kenyan government in 2014. The new collaborative health insurance program has provided healthcare coverage for all, including the Kenya’s poorest, to date. HISP is part of the World Bank Group’s Health in Africa Initiative’s support to the Kenyan government’s priority agenda of achieving universal health coverage by expanding Kenyans’ medical care coverage to the poorest and vulnerable populations [94]. The success of the pilot phase of the initiative enabled the government and its partners to scale up the program to benefit nine million poor and vulnerable Kenyans as part of the universal healthcare coverage mandates and targets.

Second, the significant policy implications in this multi-country setting have evidence-based findings presented in this study which suggest that demographic and socio-economic factors are significant predictors of health insurance coverage. Therefore, the policy options for scaling up health insurance coverage in both countries ought to model after the concept of these factors to attain the sustainability of health insurance coverage. Our findings also point to the significant role of female education positively influencing women’s health decisions in Nigeria and South Africa. For instance, there were higher odds of being insured among individuals with increasing education status in both countries. This, therefore, is a justification of closing the gap of health inequalities and creating frameworks in promoting health equity, and fostering stringent advocacy across all stakeholders in playing a major role in the social contributing factors of health [95,96,97,98,99,100,101]. Education empowers women to look after their own health by seeking appropriate healthcare when they have an ailment. Therefore, being protected by health insurance supports them to prevent catastrophic health expenses that they would have to make out-of-pocket when they fall sick and do not have health insurance coverage [97,98], and being educated makes it likely for them to insure themselves against the unforeseen out-of-pocket costs [99,100,101].

## 5. Further Discussion: Sustaining Health Insurance Coverage via Public Health Financing in Nigeria and South Africa

Attaining universal healthcare access is one of the key development priorities and a target of Sustainable Development Goal 3 (SDG 3). However, existing studies have shown that the public health systems in high-income countries such as the United States of America have a large proportion of public participation in financing the healthcare sector, as an unabridged and fixed capacity of the country’s gross domestic product (GDP), unlike in low- and middle-income countries, where public health systems are faced with challenges [101,102]. Health is a labor-intensive sector with accumulative health technology, which has significantly increased costs. In other words, public healthcare systems are untenable owing to the specificities in healthcare sectors across different countries.

Thus, healthcare systems in both countries are mostly in an impractical settings with dismal health consequences. Several studies have looked at the challenges within the healthcare system in Nigeria [1,20] and in South Africa [7,21], and our major findings to a point were akin to the conclusions found in previous studies [103,104,105,106,107,108,109]. Some of the identified challenges facing public health systems in Nigeria and in South Africa include poor human resources for the health sector, inadequate budgetary allocation to the health sector and dearth of health financing as well as poor leadership and management within the health sector [102,103]. These difficulties account for over two-thirds of the perceived problems in the healthcare sector in Africa, including in Nigeria and South Africa. Equally, when viewed from the standpoint of the World Health Organization (WHO)’s six pillars of the healthcare system [107,108], the leading problems within the health sectors in both Nigeria and South Africa still clustered around fraud and corruption, poor leadership and governance, a lacking healthcare workforce and insufficient health service delivery and financing strategies [20,21].

At the individual level, key demographic and socioeconomic factors are also among the leading challenges in the health sector in Nigeria and South Africa, as well as in other African countries [72,73,74]. Consequently, the global economic crisis and the unforeseen COVID-19 pandemic have brought an unprecedented consideration to the issue of health system sustainability in Nigeria and South Africa. Previously, both countries adopted the emerging two major types of public health systems instituted in the 1970s, named after their political instigators: *Bismarck systems* (Bismarck systems are based on social insurance, with a multitude of public insurance funds, financed by employer/employee contributions, independent of healthcare provision. Countries such as Belgium, France and Germany have this system) and *Beveridge systems* (Beveridge systems are where public financing and healthcare delivery are handled within one tax-financed structure, such as the National Health Service (NHS) in the United Kingdom and in some Nordic states) [107]. These two generic types of systems have been under intense debate, with discussions focused on access, quality and cost for countries that have adopted these public health systems. In the 2000 report of the World Health Organization (WHO) on the purpose of health financing and the 2007 expansion of the definition on health financing, the main concern was about raising adequate funds, sidestepping the implications for payers and for the economy [105,106]. With the recent economic recessions, however, universal health insurance coverage, a main pillar of social cohesion and welfare, is endangered, with profound implications for equity and financial protection [102,104]. The willingness of government and non-governmental agencies to disburse the necessary funds to the health sector in developing countries has been associated with a lack of political will, fraud and corruption within the health system [20,21]. Sustainable development and sustaining health financing remain pertinent in light of social, demographic and epidemiological changes [57,58,59,60,61,62,63,64,65,66,67,68,69,70,71,72], and the incidence of financing health system viability has long been a major subject of health policy problems across the globe, and in sub-Saharan African countries, such as in Nigeria and South Africa [1,2,3,4].

Financial sustainability has become a key healthcare concern in the 21st century across low-and middle-income countries. The real political, economic and ethical questions are the basis of the requisite financing, as it is only high-income countries that can afford to depend on mostly on private health insurance in spite of the serious equity issues involved [20,21]. Most developed and developing countries, however, have more or less established welfare states through taxation and labor contributions. It is in these countries that globalization is conveying growing economic disparity and ambiguity has instigated a major debate on the sustainability of health financing. Moreover, globalization has increased income inequality within countries with top income ranges absorbing a larger share of national gross domestic product (GDP) [28,43] and the catastrophe to tax income reduces the efficacy of welfare and safety nets and weakens the competitiveness of the economy [102,103,104]. This idea is predominantly key for low- and middle-income countries, which at present are developing their health systems through the little funding realized from national health financing budgetary allocation.

Lastly, employment contributions as a source of health financing are mismatched with universal health insurance coverage, quality of services and increasing life expectancy. A change concerning general taxation to meet healthcare needs can boost economic growth through increased competitiveness, and attain major non-health aims, such as equity, financial security, quality and responsiveness even during economic recessions [104,105,106,107]. Health system sustainability, as a system unprejudiced, must be financed through progressive tax systems and policies of all forms of tax revenue—as uncomfortable as it may seem, this is a reality not to be overlooked [102,103]. Moreover, political trepidations accompanying economic constraints as well as ethical considerations possibly will impose changes in health financing, especially in low-and middle-income countries. Thus, national health insurance financed through taxation should gain impetus in the pursuit of more justifiable and receptive health systems. Hence, the underdeveloped healthcare systems in Africa need radical clarifications with innovative thought to break the contemporary bottleneck in health financing and service delivery.

## 6. Strengths and Limitations

This study is based on nationally representative household surveys that reflect every locality in Nigeria and South Africa. To the best of our knowledge, this is among the first empirical studies to examine a comparative-cross sectional study of the prevalence and determinants associated with health insurance coverage in Nigeria and South Africa. The findings support the basis of stringent advocacy and capacity building that are tailored towards promoting the benefits of health insurance uptake in both countries. Furthermore, the findings provide useful insights for more rigorous investigations to present generalizable findings in sub-Saharan African countries. The findings of this study were deeply rooted in the study design; data were collected using standard methodologies, and sample sizes were similar in the two countries. There are some potential limitations, however, that need to be highlighted. One of the limitations of this study is that the nature of the cross-sectional design of the demographic health surveys made it impossible for causal inferences to be drawn from the findings of this study. Another limitation comes with self-reporting by the respondents of the surveys, due to factors such as the ability to remember, bias, or under-reporting.

## 7. Conclusions

The prevalence of health insurance coverage in Nigeria is lower than in South Africa, although both countries are still below the WHO internationally recognized standard measures of the health insurance coverage agenda. Nigeria and South Africa might not be able to achieve universal health coverage and meet the Sustainable Development Goals on health by the year 2030 if the present health insurance financing mechanisms persist. To achieve UHC via health insurance schemes, various health insurance platforms can be harmonized, such as government and private health financing schemes that would maximize the risk pools and coverage of potential subscribers to opt for their choice of health insurance. It is important to explore the underlying determining factors, such as demographic and socioeconomic factors, which play a vital role in the decision of health insurance uptake in households. Women’s education ought to be given more priority as it was found to be a strong predictor influencing increased coverage of health insurance. It might also be useful to both relevant stakeholders and policy makers to better understand the factors that influence decisions to purchase health insurance as well as its associated coverage complexity. Policy makers may wish to monitor developments to ensure wide coverage of health insurance in concurrence with national health policy goals in Nigeria and South Africa. In transforming health financing in both countries, it is imperative to implement programmes that will increase equity and access to healthcare services, especially among women in deprived socioeconomic households, the unemployed and vulnerable individuals.

Furthermore, other policy recommendations based on the dissimilarities found in the study are offered. The first recommendation is designated to Nigeria, as the Nigerian government is making an effort to introduce new reforms of the NHIS, which can facilitate better cooperation and harmonization of public and private health sectors. This will aid defenseless individuals such as informal employees, women with low education living in rural communities and the poorest households to access healthcare services through available and cheaper or no-cost health insurance premiums. The second recommendation, designated to South Africa, is that the South African government and other relevant health policy stakeholders should manage the racial gap in accessing healthcare services and bridging the unequal nature of healthcare provision in both public and private healthcare system across all provinces as well as strive for equal allocation of publicly funded healthcare in both rural and urban provinces/communities. Policy makers and government agencies should make efforts in implementing effective and long-term policy reforms of the institutional frameworks that allow the perpetuation of inequality in healthcare accessibility. This will further address inequities in health insurance coverage among marginalized populations, including informal employees, households with high poverty levels, women with low education, unemployed persons and rural residents.

## Figures and Tables

**Figure 1 ijerph-19-01766-f001:**
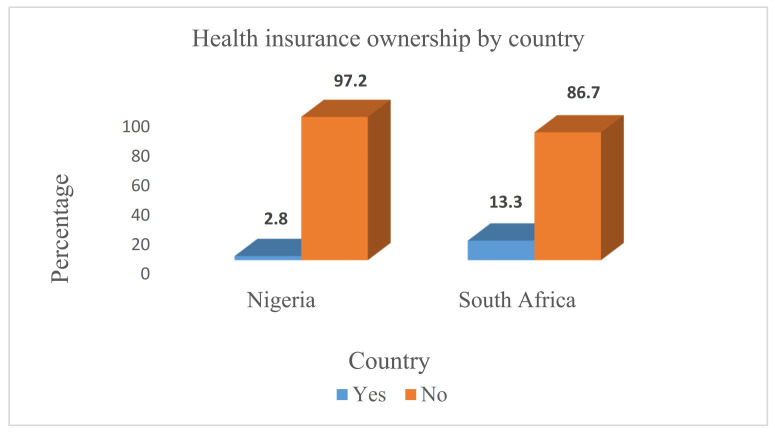
Distribution of health insurance ownership in Nigeria and South Africa.

**Figure 2 ijerph-19-01766-f002:**
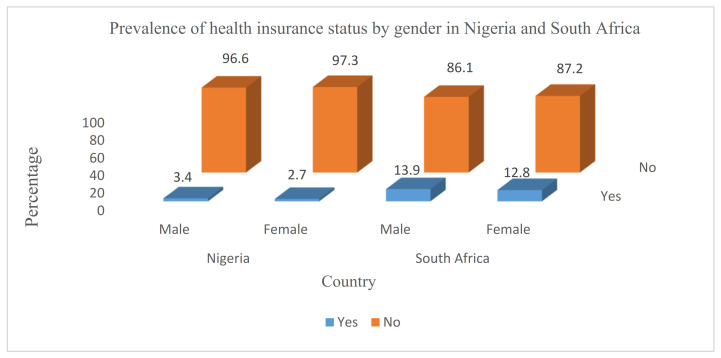
Distribution of health insurance status by gender in Nigeria and South Africa.

**Table 1 ijerph-19-01766-t001:** Geographical and socio-economic comparison of study settings.

Description	Nigeria	South Africa
**Region**	Western Africa	Southern Africa
**Topography**	Coastal with varied landscape	Coastal high and low lands
**Total land area**	923,769 km^2^	1,213,090 km^2^
**Total population**	211,400,708	60,093,707
**% rural population**	48.04%	33.3%
**% urban population**	51.96%	66.7%
**Labor force categories**	Mainly engaged in professional labor (federal, state and local government, ministries, departments and agencies). Few are engaged in non-professional labor (technical, skilled manual, unskilled manual and agriculture).	Few are engaged in formal sector (non-agricultural). Mainly engaged in informal sector (non-agricultural), agriculture and private households.
**Country unemployment rate**	32.5%	32.6%
**Country employment rate**	66.7%	38.0%
**Labor force participation rate**	53.41%	56.4%
**% of age dependency ratio**	86.7%	52.23%
**NHIS coverage (%) by gender** **Women** **Men**	3% 3%	16% 17%

**Source:** Adapted from Duku SKO (2018) [31].

**Table 2 ijerph-19-01766-t002:** Distribution of the population by socio-demographic characteristics, stratified by sex in South Africa and Nigeria.

Variables	Nigeria			South Africa		
	All No. (%)	Men No. (%)	Women No. (%)	All No. (%)	Men No. (%)	Women No. (%)
**Age**						
15–24	19,286 (35.0)	4019 (30.2)	15,267 (36.5)	4220 (34.8)	1307 (36.1)	2913 (34.2)
25–34	16,569 (30.1)	3369 (25.3)	13,200 (31.6)	3620 (29.8)	928 (25.7)	2692 (31.6)
35–44	12,751 (23.1)	3288 (24.7)	9463 (22.6)	2670 (22.0)	674 (18.6)	1996 (23.4)
45–59	6526 (11.8)	2635 (19.8)	3891 (9.3)	1622 (13.4)	709 (19.6)	913 (10.7)
**Education**						
No education	17,344 (31.5)	2946 (22.1)	14,398 (34.4)	324 (2.7)	134 (3.7)	190 (2.2)
Primary	8297 (15.0)	1914 (14.4)	6383 (15.3)	1467 (12.1)	605 (16.7)	862 (10.1)
Secondary	22,898 (41.5)	6200 (46.6)	16,698 (39.9)	9129 (75.2)	2548 (70.4)	6581 (77.3)
Higher	6593 (12.0)	2251 (16.9)	4342 (10.4)	1212 (10.0)	331 (9.2)	881 (10.4)
**Place of residence**						
Urban	22,690 (40.8)	5506 (41.4)	16,984 (40.6)	6826 (56.3)	2021 (55.9)	4805 (56.4)
Rural	32,642 (59.2)	7805 (58.6)	24,837 (59.4)	5306 (43.7)	1597 (44.1)	3709 (43.6)
**Wealth quintile**						
Poor	20,967 (38.0)	4874 (36.6)	16,093 (38.5)	5230 (43.1)	1602 (44.3)	3628 (42.6)
Middle	11,717 (21.3)	2858 (21.5)	8859 (21.2)	2800 (23.1)	844 (23.3)	1956 (23.0)
Rich	22,448 (40.7)	5579 (41.9)	16,869 (40.3)	4102 (33.8)	1172 (32.4)	2930 (34.4)
**Marital status**						
Never married	15,774 (28.6)	5105 (38.4)	10,669 (25.5)	7375 (60.8)	2241 (61.9)	5134 (60.3)
Currently married	36,906 (66.9)	8018 (60.2)	28,888 (69.1)	4035 (33.3)	1194 (33.0)	2841 (33.4)
Previously married	2452 (4.5)	188 (1.4)	2264 (5.4)	722 (5.9)	183 (5.1)	539 (6.3)
**Employment status**						
Unemployed	16,508 (29.9)	1742 (13.1)	14,766 (35.3)	7735 (63.8)	1961 (54.2)	5774 (67.8)
Employed	38,624 (70.1)	11,569 (86.9)	27,055 (64.7)	4397 (36.2)	1657 (45.8)	2740 (32.2)
**Region**						
North Central	10,187 (18.5)	2415 (18.1)	7772 (18.6)	-	-	-
North East	10,086 (18.3)	2447 (18.4)	7639 (18.3)	-	-	-
North West	13,089 (23.7)	2960 (22.2)	10,129 (24.2)	-	-	-
South East	7326 (13.3)	1755 (13.2)	5571 (13.3)	-	-	-
South South	6777 (12.3)	1697 (12.8)	5080 (12.1)	-	-	-
South West	7667 (13.9)	2037 (15.3)	5630 (13.5)	-	-	-
**Province**						
Western Cape	-	-	-	876 (7.2)	220 (6.1)	656 (7.7)
Eastern Cape	-	-	-	1516 (12.5)	475 (13.1)	1041 (12.2)
Northern Cape	-	-	-	1017 (8.4)	299 (8.3)	718 (8.4)
Free State	-	-	-	1190 (9.8)	336 (9.3)	854 (10.0)
KwaZulu-Natal	-	-	-	1884 (15.5)	524 (14.5)	1360 (16.0)
North West	-	-	-	1291 (10.6)	428 (11.8)	863 (10.1)
Gauteng	-	-	-	1279 (10.5)	416 (11.5)	863 (10.1)
Mpumalanga	-	-	-	1519 (12.5)	465 (12.8)	1054 (12.4)
Limpopo	-	-	-	1560 (12.9)	455 (12.6)	1105 (12.0)
**Population group**						
Black/African	-	-	-	10,509 (86.6)	3150 (87.1)	7359 (86.4)
Other	-	-	-	1623 (13.4)	468 (12.9)	1155 (13.6)

Significant *p*-values: *p* < 0.05; *p* < 0.001; 95% confidence intervals (CI).

**Table 3 ijerph-19-01766-t003:** Distribution of health insurance coverage by socio-demographic characteristics in South Africa and Nigeria.

Variables	Nigeria			South Africa		
	No (%)	Yes (%)	*p*-Value	No. (%)	Yes (%)	*p*-Value
**Gender**			0.132			0.145
Male	12,861 (24.0)	450 (28.7)		3114 (46.0)	504 (48.4)	
Female	40,704 (76.0)	1117 (71.3)		3656 (54.0)	537 (51.6)	
**Age**			0.000 *			0.000 *
15–24	18,984 (35.4)	302 (19.3)		2479 (36.6)	244 (23.4)	
25–34	16,113 (30.1)	456 (29.1)		1988 (29.4)	274 (26.3)	
35–44	12,205 (22.8)	546 (34.8)		1372 (20.3)	291 (28.0)	
45–59	6263 (11.7)	263 (16.8)		931 (13.7)	232 (22.3)	
**Education**			0.000 *			0.000 *
No education	17,230 (32.2)	114 (7.3)		221 (3.3)	9 (0.9)	
Primary	8237 (15.4)	60 (3.8)		979 (14.5)	48 (4.6)	
Secondary	22,407 (41.8)	491 (31.3)		3140 (75.9)	667 (64.1)	
Higher	5691 (10.6)	902 (57.6)		430 (6.3)	317 (30.4)	
**Place of residence**			0.000 *			0.000 *
Urban	21,413 (40.0)	1077 (68.7)		3603 (53.2)	772 (74.2)	
Rural	32,152 (60.0)	490 (31.3)		3167 (46.8)	269 (25.8)	
**Wealth index**			0.000 *			0.000 *
Poor	20,875 (39.0)	92 (5.9)		3238 (47.8)	133 (12.8)	
Middle	11,584 (21.6)	133 (8.5)		1670 (24.7)	155 (14.9)	
Rich	21,106 (39.4)	1342 (85.6)		1862 (27.5)	753 (72.3)	
**Marital status**			0.000 *			0.000 *
Never married	15,403 (28.8)	371 (23.7)		4365 (64.5)	449 (43.1)	
Currently married	35,758 (66.7)	1148 (73.2)		2009 (29.7)	546 (52.5)	
Previously married	2404 (4.5)	48 (3.1)		396 (5.8)	46 (4.4)	
**Employment status**			0.000 *			0.000 *
Unemployed	16,142 (30.1)	366 (23.4)		4465 (65.9)	362 (34.8)	
Employed	37,423 (69.9)	1201 (76.6)		2305 (34.1)	679 (65.2)	
**Region**			0.000 *			
North Central	9777 (18.3)	410 (26.2)		-	-	
North East	9969 (18.6)	117 (7.5)		-	-	
North West	12,712 (23.7)	377 (24.1)		-	-	
South East	7135 (13.3)	191 (12.2)		-	-	
South South	6599 (12.3)	178 (11.4)		-	-	
South West	7373 (13.8)	294 (18.7)		-	-	
**Province**						0.000 *
Western Cape	-	-		373 (5.5)	143 (13.7)	
Eastern Cape	-	-		875 (12.9)	108 (10.4)	
Northern Cape	-	-		563 (8.3)	99 (9.5)	
Free State	-	-		670 (9.9)	87 (8.4)	
KwaZulu-Natal	-	-		1089 (16.1)	126 (12.1)	
North West	-	-		705 (10.4)	121 (11.6)	
Gauteng	-	-		701 (10.4)	139 (13.4)	
Mpumalanga	-	-		889 (13.1)	99 (9.5)	
Limpopo	-	-		905 (13.4)	119 (11.4)	
**Population group**						0.000 *
Black African	-	-		6078 (89.8)	723 (69.4)	
Other	-	-		692 (10.2)	318 (30.6)	

Significant *p*-values: *p* < 0.05; 95% confidence intervals (CI), “*” stands for the *p*-value explaining the significance (*p* < 0.05) of the variables.

**Table 4 ijerph-19-01766-t004:** Logistic regression analysis identifying associations between socio-demographic factors and health insurance coverage in Nigeria and South Africa.

	Nigeria		South Africa	
	Model 1 (uOR)	Model 2 (aOR)	Model 1 (uOR)	Model 2 (aOR)
	OR (95% CI)	OR (95% CI)	OR (95% CI)	OR (95% CI)
**Gender**				
Male	RC	RC	RC	RC
Female	1.28 (1.15–1.43) *	1.85 (0.75–1.97) *	1.06 (0.93–1.20) *	1.91 (0.78–2.05) *
**Age**				
15–24	RC	RC	RC	RC
25–34	0.53 (0.46–0.62) *	0.80 (0.67–0.96) *	0.76 (0.64–0.90) *	1.82 (1.46–1.98) *
35–44	0.36 (0.31–0.42) *	0.49 (0.40–0.60) *	0.52 (0.43–0.61) *	1.29 (1.02–1.63) *
45–59	0.36 (0.30–0.42) *	0.40 (0.32–0.50) *	0.36 (0.30–0.43) *	1.01 (0.78–1.32) *
**Education**				
No education	RC	RC	RC	RC
Primary	1.07 (0.78–1.09) *	1.05 (0.04–1.20) *	1.05 (0.03–1.11) *	1.01 (0.05–1.24) *
Secondary	1.08 (0.06–1.10) *	1.09 (0.95–1.88) *	1.06 (0.05–1.28) *	1.02 (0.15–1.67) *
Higher	1.43 (0.34–1.54) *	1.34 (0.28–1.42) *	1.33 (0.16–1.66) *	1.76 (0.34–1.82) *
**Place of residence**				
Urban	RC	RC	RC	RC
Rural	3.23 (2.89–3.63) *	1.07 (0.94–1.23) *	2.52 (2.15–2.95) *	0.60 (0.47–0.77) *
**Region**				
North Central	RC	RC	-	-
North East	2.14 (1.72–2.67) *	1.21 (0.96–1.52)	-	-
North West	1.10 (0.94–1.30)	0.59 (0.49–0.70) *	-	-
South East	1.01 (0.83–1.23)	1.56 (1.28–1.92) *	-	-
South South	1.08 (0.89–1.32)	1.75 (1.42–2.16) *	-	-
South West	0.74 (0.63–0.87) *	1.55 (1.30–1.85) *	-	-
**Province**				
Western Cape	-	-	RC	RC
Eastern Cape	-	-	3.26 (2.52–4.21) *	0.84 (0.61–1.16)
Northern Cape	-	-	2.53 (1.58–4.05) *	1.26 (0.74–2.16)
Free State	-	-	3.24 (2.29–4.57) *	1.07 (0.72–1.61)
KwaZulu-Natal	-	-	3.36 (2.68–4.20) *	1.06 (0.80–1.40)
North West	-	-	2.12 (1.61–2.78) *	0.53 (0.37–0.75) *
Gauteng	-	-	2.13 (1.77–2.57) *	0.93 (0.73–1.22)
Mpumalanga	-	-	3.77 (2.79–5.10) *	1.08 (0.74–1.57)
Limpopo	-	-	3.02 (2.32–3.94) *	0.63 (0.44–0.89) *
**Population group**				
African/Black	-	-	RC	RC
Others	-	-	1.19 (0.16–1.21) *	1.43 (0.35–1.53) *
**Wealth quintile**				
Poor	RC	RC	RC	RC
Middle	0.49 (0.38–0.65) *	0.65 (0.49–0.87) *	0.34 (0.26–0.44) *	0.40 (0.31–0.53) *
Rich	1.09 (0.07–1.11) *	1.20 (0.15–1.25) *	1.07 (0.06–1.10) *	1.10 (0.08–1.13) *
**Marital status**				
Never married	RC	RC	RC	RC
Currently married	1.11 (0.82–1.50) *	1.20 (0.86–1.68) *	1.83 (0.61–1.98) *	1.05 (0.74–1.51) *
Previously married	0.74 (0.65–0.83)	0.74 (0.63–0.87)	0.38 (0.33-.043)	0.53 (0.44–0.63)
**Employment status**				
Unemployed	RC	RC	RC	RC
Employed	1.69 (0.61–1.78) *	1.05 (0.91–1.21)	1.24 (0.21–1.37) *	1.33 (0.28–1.50) *

Significant *p*-value: *p* < 0.05; 95% confidence interval (CI); RC = reference category; uOR = unadjusted odds ratio; aOR = adjusted odds ratio, “*” stands for the p-value explaining the significance (*p* < 0.05) of the variables.

## Data Availability

The datasets analyzed during the current study are available from the DHS Program: https://dhsprogram.com/data/available-datasets.cfm. (accessed on 15 May 2021).

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
