# Peer review of "A Comparative Cross-Sectional Study of the Prevalence and Determinants of Health Insurance Coverage in Nigeria and South Africa: A Multi-Country Analysis of Demographic Health Surveys"

_ijerph, 2022, doi:10.3390/ijerph19031766_

Round 1

Reviewer 1 Report

Dear authors, 

thank you  for letting me read your paper. Follows some comments over the paper. I hope I helped you.

1- Why were Nigeria and South Africa selected? What is the association between two countries?

It is necessary to insert a justificative short in abstract, it is need explain why the paper is analyze these countries. It is need to insert in main text the justificative too.

2- The author(s) could improve the discussion of the public health system, because in developed countries, with the exception of the USA, all health systems have a large percentage of public participation, whether in financing, organization of the system as a whole and own installed capacity. This is because health is a labor-intensive sector with accumulative technology, which greatly increases its costs, with a growing profile. In other words, private health systems are unsustainable due to the specificities of the health sector. In other words, in low- and middle-income countries, the challenge is even greater. Thus, the authors could discuss the possibilities of public systems in these countries.

For example, discussing the type of financing for the health system with progressive taxation, as in developed countries. Reducing income and wealth inequality and redistributing it through quality public services such as healthcare.

Author Response

Dear Reviewer 1, 

Thank you for reading our manuscript and all your comments/suggestions were welcome and effected in main text of the manuscript in blue colour ink. This has helped our manuscript greatly.

Thank you once again.

Best regards,

Monica Akokuwebe

Responses to Reviewer 1 Comments

1- Reviewer Comment: Why were Nigeria and South Africa selected? What is the association between two countries?

Author(s) Response(s): This was done in 2.2 as justification of the study setting in Page 5

  1. Reviewer Comment: It is necessary to insert a justificative short in abstract, it is need explain why the paper is analyze these countries. It is need to insert in main text the justificative too.

Author(s) Response(s): The corrections has been made on the Abstract and this was done on page 1

3- Reviewer Comment: The author(s) could improve the discussion of the public health system, because in developed countries, with the exception of the USA, all health systems have a large percentage of public participation, whether in financing, organization of the system as a whole and own installed capacity. This is because health is a labor-intensive sector with accumulative technology, which greatly increases its costs, with a growing profile. In other words, private health systems are unsustainable due to the specificities of the health sector. In other words, in low- and middle-income countries, the challenge is even greater. Thus, the authors could discuss the possibilities of public systems in these countries. For example, discussing the type of financing for the health system with progressive taxation, as in developed countries. Reducing income and wealth inequality and redistributing it through quality public services such as healthcare.

Author(s) Response(s): The corrections has been made on the main text of the manuscript on pages 19 to 21.

Reviewer 2 Report

This paper does not provide new evidence on factors affecting health insurance coverage, but confirms prior studies. The focus on two non-randomly-selected countries and data for two different years, is unusual and not well-motivated. Most of the paper is descriptive, which may be valuable if there are not other reports containing these statistics. It is not clear what the authors are trying to test by comparing statistics in teh two countries. The authors should discuss why they might expect their covariates to have different effects in the two countries and, in the discussion of the results (e.g., on page 15), provide more discussion about why estimated probabilities differ across the two models. Did the authors consider running a pooled regression with interaction terms?  

The study would be strengthened if the authors put more emphasis on comparing and contrasting. The "obvious" need to adopt the Kenya NHIP is not well-supported by the analysis: the authors could better motivate why they think this strategiy would using their statistical results. E.g., this is a weak recommendation (page 17): "the evidence presented in this study suggests that demographic and socioeconomic factors are significant predictors of health insurance coverage. Therefore, the policy options for scaling-up health insurance coverage in both countries ought to model on the concept of these factors to attain the sustainability of health insurance coverage."

If the results are different for the two countries, wouldn't policy recommendations differ? I do not feel like there is adequate attention to the differences.

Author Response

Dear Reviewer 2, 

Thank you for reading our manuscript and all your comments/suggestions were welcome and effected in main text of the manuscript in green colour ink.

This has helped our manuscript greatly.

Thank you once again.

Best regards,

Monica Akokuwebe

Responses to Reviewer 2 Comments/Suggestions

Reviewer 2: This paper does not provide new evidence on factors affecting health insurance coverage, but confirms prior studies. The focus on two non-randomly-selected countries and data for two different years, is unusual and not well-motivated. Most of the paper is descriptive, which may be valuable if there are no other reports containing these statistics. It is not clear what the authors are trying to test by comparing statistics in the two countries. The authors should discuss why they might expect their covariates to have different effects in the two countries and, in the discussion of the results (e.g., on page 15), provide more discussion about why estimated probabilities differ across the two models. Did the authors consider running a pooled regression with interaction terms? 

Author(s) Responses: First, this paper provided new evidence on factors affecting health insurance coverage as the data (2018  NDHS and 2016 SADHS) has all the variables (factors) that this study examine. And you must look at our objectives and what we intend to seek for. The use of 2018 NDHS and 2016 SADHS is usual and well-motivated for this study as this is a national representative data. One thing you must understand about this study is that national representative data are collected and researcher(s) make(s) use of the data to draw inferences from the data to compliment the literature gotten from previous studies. We compared the findings gotten from the analysed data to draw comparisons between the two countries.

Page 15: Adding covariates reduces the bias in your predictions, but increases the variance. This showed that adding covariates can greatly improve the accuracy of the model and may significantly affect the final analysis results. Including a covariate in the model can reduce the error in the model to increase the power of the fact tests. Also, similar to an independent variable, a covariate is complementary to the dependent or response variable. So the independent variable is measurable and considered to have a statistical relationship with the outcome variable which qualify the independent as a potential covariate. Thus, adjustment for such covariates generally improves the efficiency of the analysis and hence, produces stronger and more precise evidence (smaller ρ-values and narrower confidence intervals) of an effect.

The covariates findings must have different effects in the two countries and in the discussion of the results as we analysed the two different DHS data separately (2018 NDHS and 2016 SADHS) and take a good look by reading with interest the methodology to see the characteristics of the population in both countries. Also, the sample size of the respondents for each country were different and also remember the sample size can increase precision of the covariates for each country.

Estimated probabilities must differ across the two models as the DHS data are different and the sample size varies too with different characteristics of the population.

Data pooling is a process where data sets are coming from different sources are combined. This can mean two things: first, that multiple datasets containing information on many respondents from different countries or from different institutions is merged into one data file. Second, that data on one country, coming from multiple sources such as e.g. primary care, specialist clinics etc are combined together. Yes, in both cases pooling results in a fuller and more useful dataset for scientific research. The interaction effects will occur when the effect of one variable depends on the value of another variable. First, you must look at the topic and then the aim of the study. We want to see the comparisons between South Africa and Nigeria on their operations of health insurance coverage, hence pooling the data will not be feasible with this objective. Pooled data goes with time series and more data are merged together to look at a particular trend with a geographical settings. For instance, I can pooled a data from 32 countries across West Africa, East Africa, Southern Africa and Central Africa from different period of time and analyse for trends across these 32 countries merged as “Pooled data of 32 countries examining health insurance prevalence and coverages across sub-Saharan Africa”…….this can be employed and the findings from the pooled data will offer better interaction effects. So, in this study we wanted to look at two countries in a separate and differently. When you merge or pool data together, you cannot operate comparative analysis anymore.

Reviewer 2: The study would be strengthened if the authors put more emphasis on comparing and contrasting. The "obvious" need to adopt the Kenya NHIP is not well-supported by the analysis: the authors could better motivate why they think this strategy would using their statistical results. E.g., this is a weak recommendation (page 17): "the evidence presented in this study suggests that demographic and socioeconomic factors are significant predictors of health insurance coverage. Therefore, the policy options for scaling-up health insurance coverage in both countries ought to model on the concept of these factors to attain the sustainability of health insurance coverage."

Author(s) response(s): This has been addressed in pages 18 to 19.

Reviewer 2: If the results are different for the two countries, wouldn't policy recommendations differ? I do not feel like there is adequate attention to the differences.

Author(s) response(s): This has been addressed in pages 21 to 22.

Round 2

Reviewer 2 Report

Revisions are good. I have no more concerns.